# Retrosynthetic planning with experience-guided Monte Carlo tree search

Siqi Hong[1], Hankz Hankui Zhuo [1✉], Kebing Jin[1], Guang Shao[2] & Zhanwen Zhou[1]

In retrosynthetic planning, the huge number of possible routes to synthesize a complex molecule using simple building blocks leads to a combinatorial explosion of possibilities. Even experienced chemists often have difficulty to select the most promising transformations. The current approaches rely on human-defined or machine-trained score functions which have limited chemical knowledge or use expensive estimation methods for guiding. Here we propose an experience-guided Monte Carlo tree search (EG-MCTS) to deal with this problem. Instead of rollout, we build an experience guidance network to learn knowledge from synthetic experiences during the search. Experiments on benchmark USPTO datasets show that, EG-MCTS gains significant improvement over state-of-the-art approaches both in efficiency and effectiveness. In a comparative experiment with the literature, our computer-generated routes mostly matched the reported routes. Routes designed for real drug compounds exhibit the effectiveness of EG-MCTS on assisting chemists performing retrosynthetic analysis.

[1] School of Computer Science and Engineering, Sun Yat-Sen University, East Outer Ring Road, 510006 Guangzhou, Guangdong, China. [2] School of Chemistry, Sun Yat-Sen University, East Outer Ring Road, 510006 Guangzhou, Guangdong, China. ✉email: zhuohank@mail.sysu.edu.cn

Chemical synthetic analysis, i.e., retrosynthesis, aims at designing a pathway to synthesize the target molecule using a set of available building blocks[1]. Computer-assisted approaches have been an active research topic since Corey and Wipke[2] created the first computer program for retrosynthetic planning, after which great progress[3–10] has been made with the development of large reaction databases[11]. The retrosynthetic task is challenging since the search space of available reactions in each step is prohibitively large.

There have been approaches on single-step retrosynthesis, such as template-based[12–14] and template-free[15–20] approaches, which aim to predict all promising single-step decompositions for the target molecule. Based on single-step retrosynthesis, in this paper we investigate the problem of multi-step retrosynthesis, which is challenging since we need to consider various combinations of substantial reactions of multiple steps. There have been approaches proposed to tackle this challenge by building score functions, which are either human-defined or machine-trained, to guide the search of reactions. The role of the score function is to assess the value of a searching state, such as predicting the cost of a molecule to be retro-synthesized or a reaction to be applied to decompose molecules. For example, 3N-MCTS[4] combined Monte Carlo Tree Search (MCTS) with three networks to perform chemical synthesis planning, using Rollout to estimate the score function of a searching state. Kishimoto et al.[6] proposed a human-defined score function to select reactions with the lowest cost based on a depth-first proof-number search. Coley et al.[21] proposed an approach toward fully autonomous chemical synthesis that combines techniques in artificial intelligence for planning and robotics for execution. Molga et al.[22] proposed Synthia program, a commercial software platform to design synthetic pathways. Klucznik et al.[23] executed the routes planned autonomously by Chematica in the laboratory and provided the validation of the computer approach in synthetic design. Chen et al.[7] proposed an approach, called Retro*, to do A* search of reactions with the guidance of previously trained neural network which predicts the synthetic cost of the molecule. Recently, Kim et al.[10] proposed a self-improving procedure to enhance the existing approaches, such as Retro*. We call this enhanced approach Retro*+ for simplicity. Reinforcement learning based approaches[5,9] were also proposed to build score functions with the similarity of the retrosynthetic problems to strategy games[24].

Despite the success of previous approaches, the learning components they relied on are often based on existing single-step reaction databases (such as USPTO[11]), such as the three networks in 3N-MCTS[4], the policy and value networks in Retro*[7]. The knowledge they can acquire mainly depends on the quality and quantity of the databases. More importantly, the existing databases only contain single-step reactions. It is thus difficult for current learning components to derive multi-step information and knowledge directly from them. In other words, it is challenging to build a path-level and forward-looking score function to accurately predict molecules or reactions.

Figure 1 shows the searching process for the target molecule $A$. Approaches such as Retro* guide the search by learning a score function that predicts synthetic cost of molecules. Retro* constructs multi-step synthetic routes from single-step reaction datasets. Since there are reaction $I + J \rightarrow H$ and reaction $G + H \rightarrow F$ in datasets, the score function learns that the cost of molecules $H$ and $F$ are 1 and 2, respectively. Reaction $D + E \rightarrow C$ is not, however, included in the datasets, so molecule $C$ may have a higher predictive cost than $H$ and $F$. The base algorithm of Retro*, A* search, prefers to search molecules with lower synthetic cost, resulting in selecting template $T_{A_2}$ first. Once template $T_{A_2}$ is proved to be successful, Retro*+ will further increase its

probability such that other potential better route, e.g., the one going to template $T_{A_1}$, will not be selected. In terms of route length, the route going to template $T_{A_1}$, which could be shorter than the one going to $T_{A_2}$, may not be explored by Retro*+. Based on this observation, we conjecture that leveraging all potential templates from the template library to help construct synthetic routes could be helpful for guiding retrosynthetic planning when doing the MCTS search.

Besides, we also observe that there are many experiences that fail to construct successful route to synthesize target molecules with the building blocks during self-play. For example, the synthetic route through molecule $K$ and $L$, shown in Fig. 1, is not a successful one, since $N$ does not belong to the building blocks. Those failed experiences can be used to learn score functions for guiding retrosynthetic planning without similar failures. Note that previous approaches, such as Retro*, Retro*+, and those RL-based approaches[5,9] used the learned score function to estimate the expected synthetic cost or value of any given molecule. Since Retro* is only trained from successfully synthesized molecules, the failed synthetic pathways, which could be helpful for improving the synthetic performance, are neglected. RL-based approaches learn the score function during self-play, considering failure experiences by setting penalty values (high synthesis cost or low synthesis value) for failed or unproven molecules while searching. Unlike setting penalty values, our approach aims to make the score function reflect the actual decomposition situation, especially those unproven.

Based on the above-mentioned two observations, we propose a MCTS-based search approach, namely EG-MCTS, standing for Experience-Guided Monte Carlo Tree Search, to generate routes for synthesizing target molecules. We follow the common practice to ignore the reagents and other chemical reaction conditions. We first learn an Experience Guidance Network (EGN) to estimate the score function of reaction templates by collecting retrosynthetic experiences. We then generate retrosynthetic routes for target molecules with the learnt EGN.

To explore the low-probability but potentially successful reaction templates in the template library when collecting synthetic experiences, EG-MCTS uses MCTS to explore reaction templates and records the scores of these templates for training the score function. Different from A* search, the core component of MCTS, "upper confidence bound" (UCB), balances the trade-off between exploration of infrequently-visited routes and exploitation of high-value routes. It makes the composite score of high-value routes to decrease as the number of visits increases. Even though EGN may predict a higher score for $T_{A_2}$ in the random initial stage, the search will later turn to explore $T_{A_1}$ due to the decreasing score of $T_{A_2}$ after multiple visits to $T_{A_2}$. Therefore, our EG-MCTS approach will find that template $T_{A_1}$ leads to a fewer-step route during the MCTS exploration and record the experiences about $T_{A_1}$ for future exploration. To leverage the failed experiences, we estimate the scores of reaction templates with the failed experiences along with the successful experiences. For example, in Fig. 1, the route through molecule $K$ and $L$ fails (or has not been verified) to reach a successful synthetic route. We estimate that the score of reaction template $T_{L_1}$ is 1/2, considering it breaks molecule $L$ into $M$ and $N$, where $M$ belongs to the building blocks while $N$ does not.

In conclusion, we propose EG-MCTS, a MCTS-based search approach, to deal with retrosynthetic planning problem. The experimental results demonstrate our approach gains significant improvement over existing approaches. The comparative experiment with the literature confirms the validity and feasibility of our computer-generated routes. The results of retrosynthetic planning for realistic drugs or compounds also exhibit that EG-MCTS is instructive.

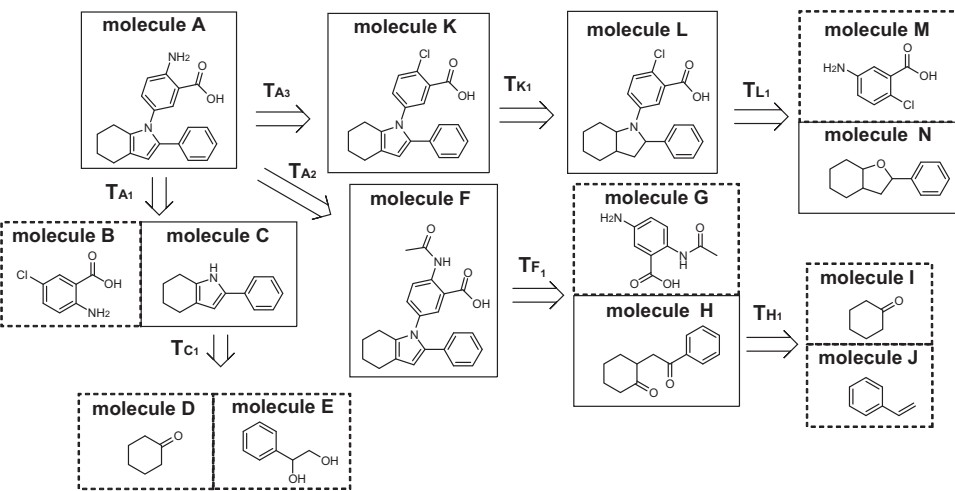

**Fig. 1 The searching process for the target molecule A.** Molecules in the dashed box belong to building blocks.

## Results and discussion

**Formulation of retrosynthetic planning**. In general, the input of retrosynthetic planning, or RS planning, is composed of a target molecule $m_0$, a building blocks set $\mathcal{B}$, and a single-step retrosynthetic model $S(\cdot)$. $\mathcal{B}$ is composed of a set of simple, commercially available molecules. A single-step retrosynthetic model $S(\cdot)$ takes a molecule $m$ as input, predicts $k$ reaction templates $T$ with the highest probability, and outputs their probabilities $P$ as well. It can be formulated as $S(m) : \{T_j, P(m, T_j)\}_{j=1}^{k}$, where $P(m, T_j)$ indicates the probability $j$th template $T_j$ given molecule $m$. There have been off-the-shelf approaches[3,4,7,12] developed to build this model effectively. In this work, we borrow the single-step model developed by Kim et al.[10]. The output of RS planning is a synthetic route from $\mathcal{B}$ to $m_0$, i.e., a series of chemical reactions whose reactants are directly from $\mathcal{B}$ or synthesized from $\mathcal{B}$.

**EG-MCTS overview**. Our EG-MCTS approach is composed of two phases, i.e., (I) learning an Experience Guidance Network (EGN) for guiding the search, and (II) generating synthetic routes for molecules with the learnt EGN (shown in Fig. 2a).

In order to deal with the difficulty in defining a score function and the lack of path-level synthetic routes for learning, in Phase I we aim to use a network-guided MCTS planning to collect synthetic experience, and then use the experience to update the network. Monte Carlo Tree Search[25] as a general search approach, has been demonstrated successful in games, such as Go[26–28]. A variant of MCTS, PUCT[29], has been successfully applied for RS planning[4]. We use a neural network instead of the traditional Rollout strategy to calculate heuristic values of searching nodes. This network, namely Experience Guidance Network, estimates a score $Q$ for each template $T$ acting on each molecule $m$ as the initial evaluation value.

In Phase I shown in Fig. 2a, we first initialize the EGN with random weights. For each target molecule in training set, we build a search tree using EG-MCTS planning with EGN and collect the synthetic experience based on the search tree as the training data of EGN. Then we update the EGN. After getting the new EGN, we verify its performance on the validation set. If it reaches the optimal performance, Phase I stops and returns the well-trained EGN. Otherwise, the Phase I will loop in the order of experience collecting, EGN updating and EGN validating.

So far we have obtained the well-trained EGN from Phase I and in Phase II, we use it to guide EG-MCTS planning. After generating the search tree for a new target molecule, we analyze the synthetic routes from the tree.

The key part, EG-MCTS planning appears in both Phase I and II, helping to collect synthetic experience and generate the synthetic routes. The search tree built by EG-MCTS planning is represented as an AND-OR tree. The OR node (molecule node) contains a molecule and the AND node (reaction node) contains a reaction template. The planning procedure can be found from Fig. 2b, which is composed of three modules, i.e., Selection, Expansion and Update. The Selection module selects the most promising molecule node $m$, and the Expansion module expands the selected node using the single-step retrosynthetic model and predicts the initial value using EGN. After that, the Update module updates upwards along the tree. These three molecule modules loop continuously until the search cost is exhausted.

**Experimental results**. We evaluated EG-MCTS with comparison to baseline approaches in the test set of 180 molecules we collect and the test set of Retro*[7] and Retro*+[10], called Retro*-190. The building blocks set $\mathcal{B}$ comes from eMolecules. We considered the first route of each molecule generated by all approaches to calculate the evaluation metrics, under the assumption that a good algorithm should be able to find paths of high-quality as quickly as possible in practice, as done by Retro*[7], Retro*+[10], and DFPN-E[6]. Our evaluation metrics include the efficiency of the planning and the quality of the solution routes. We also evaluated the bias of our approach towards the search method.

For the efficiency of planning, since the call of $S(\cdot)$ occupies most of running time, and there is always a model call in every iteration, we use the average number of iterations (Avg iter) as a measure of time and we compare the success rate of all approaches under the same iteration limit, referred to others[6,7,10]. We also compare the average number of molecule nodes (Avg M) and reaction nodes (Avg T) expanded by the various approaches during the searching processes.

Table 1 shows the planning efficiency performance of all approaches on our test set and Retro*-190, respectively. The metrics, Avg iter, Avg T, and Avg M are under the iteration limit of 500. With the assistance of our EGN, the performance of EG-MCTS is much better than the non-learning version in all metrics, demonstrating the performance improvement brought by our EGN. EG-MCTS is 3.88% more successful than the sub-optimal approach, Retro*+ and uses 25.22 fewer iterations than Retro*+ in our test set. In Retro*-190, our EG-MCTS also has a great advantage in the metric avg iter. The success rate of iter limit of the Table 1 show the effect of iteration limit on the success rate of these algorithms. We can see that our EG-MCTS

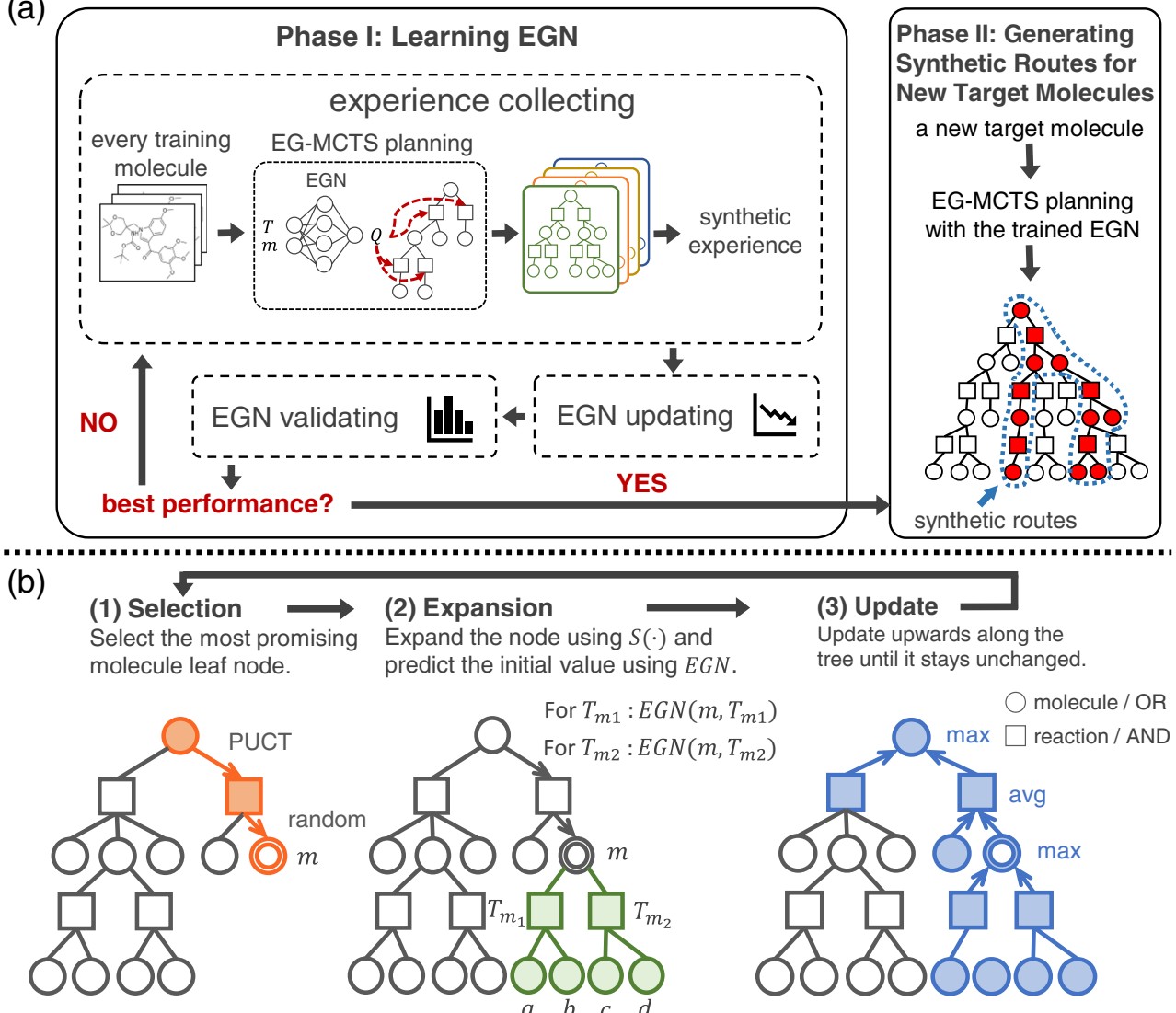

**Fig. 2 Overview of experience-guided Monte Carlo tree search approach (EG-MCTS) and the procedure of the key part, EG-MCTS planning. a** Two phases of EG-MCTS approach. Phase I is to learn the Experience Guidance Network (EGN) and Phase II is to generate synthetic routes for new target molecules. **b** Three modules of the EG-MCTS planning procedure. Selection, expansion and update are executed in a loop until the search cost is exhausted. "circles" and "squares" indicate molecule nodes and reaction nodes, respectively. "Double circles" indicate the molecule nodes are selected by the Selection module and the path marked orange shows the Selection process. Those nodes marked green are expanded by the Expansion module, and the blue path shows the Update process.

performs super well at the beginning on both two test sets. These phenomena indicate that our collected experience through self-play is of better quality and more instructive. The EGN can help the search to focus on more promising actions and to avoid entering a hopeless path so that accelerate the searching process.

We explore the performance of EG-MCTS and Retro*+ under the iteration limit of 5000. The result shows that the success rates of both algorithms converge to the same value (98.42% in Retro*-190 and 96.11% in our test set), while the average number of iteration of EG-MCTS remains less than Retro*+. Theoretically, if we do not limit the search cost, any search algorithm can find the solution for a target molecule which can be solved.

Except Greedy DFS, there are 132 molecules successfully solved by all approaches on our test set and 103 molecules successfully solved on Retro*-190. To measure the quality of the solution routes, we compare the route length, that is the number of reactions in the route. The results are shown in Table 2. The metric LRN (number of longest routes) of an approach indicates

the number of longest routes generated by the approach over all of the successfully solved molecules. Specifically, for each molecule successfully solved by all approaches, if an approach generates the longest route over all approaches, the LRN of this approach is increased by one. Similarly, the metric SRN (number of shortest routes) of an approach indicates the number of shortest routes generated by the approach over all of the successfully solved molecules. The metric Avg indicates an average of length of all routes generated by each approach.

Our approach has superior comprehensive performance among all approaches, showing the guiding role of our EGN in finding high-quality routes. Although Retro*+ and Retro*-0+ perform well in planning efficiency, but the quality of the routes they give is not so good on both two test sets. We consider the reason may be that when performing self-improvement, they simply increase the probability of those paths which have been proven successful. In our EG-MCTS, we learn a a comprehensive score for the path, so we can fully consider all potential paths.

**Table 1 Planning efficiency performance on our test set of 180 molecules and Retro*-190.**

| Algorithm | Success rate of iter limit(%) | | | | | Avg iter | Avg T | Avg M |
|---|---|---|---|---|---|---|---|---|
| | 100 | 200 | 300 | 400 | 500 | | | |
| Performance on our test set of 180 molecules | | | | | | | | |
| EG-MCTS | **85.00** | **90.00** | **92.78** | **93.33** | **94.44** | **60.75** | **837.56** | **1133.90** |
| EG-MCTS-0 | 77.78 | 78.89 | 80.56 | 80.56 | 81.11 | 128.96 | 1411.80 | 1904.21 |
| Retro*+ | 81.11 | 85.56 | 86.67 | 87.22 | 90.56 | 85.97 | 927.46 | 1396.27 |
| Retro*-0+ | 80.56 | 82.78 | 86.67 | 86.675 | 89.44 | 87.87 | 1056.01 | 1612.05 |
| MCTS-rollout | 73.33 | 77.78 | 74.21 | 74.21 | 78.89 | 133.69 | – | – |
| DFPN-E | 56.11 | 62.22 | 68.89 | 72.22 | 76.67 | 170.34 | 2271.56 | 3012.49 |
| Greedy DFS | 45.00 | 48.89 | 50.00 | 51.11 | 54.44 | 268.59 | – | – |
| Performance on test set Retro*-190 | | | | | | | | |
| EG-MCTS | **85.79** | **92.63** | **94.21** | **95.79** | **96.84** | **55.84** | **869.59** | **1193.79** |
| EG-MCTS-0 | 57.37 | 63.68 | 68.42 | 71.05 | 73.68 | 186.15 | 2525.20 | 3339.52 |
| Retro*+ | 71.05 | 85.26 | 88.95 | 90.00 | 91.05 | 100.15 | 1209.79 | 1767.81 |
| Retro*-0+ | 67.37 | 82.10 | 93.16 | 95.26 | 96.32 | 96.14 | 1421.90 | 2108.50 |
| MCTS-rollout | 43.68 | 47.37 | 54.74 | 58.95 | 62.63 | 254.32 | – | – |
| DFPN-E | 50.53 | 58.42 | 64.21 | 68.42 | 75.26 | 208.12 | 3123.33 | 4635.08 |
| Greedy DFS | 38.42 | 40.53 | 44.21 | 45.26 | 46.84 | 300.56 | – | – |

The metric Avg iter is the average number of iterations. The metrics Avg T and Avg M are the average number of reaction nodes and molecule nodes expanded by the various approaches during the searching processes.
Values marked with bold are the best performance under each metric.

**Table 2 Route quality performance on 132 molecules successfully solved on our test set and 103 molecules successfully solved on Retro*-190.**

| Algorithm | our test set | | | Retro*-190 | | |
|---|---|---|---|---|---|---|
| | LRN | SRN | Avg | LRN | SR | Avg |
| EG-MCTS | **7** | **117** | **5.85** | **13** | **51** | **5.07** |
| EG-MCTS-0 | 90 | 20 | 8.15 | 20 | 23 | 5.87 |
| Retro*+ | 96 | 12 | 8.37 | 26 | 24 | 6.03 |
| Retro*-0+ | 104 | 10 | 8.48 | 40 | 24 | 6.25 |
| MCTS-rollout | 98 | 13 | 8.23 | 30 | 26 | 6.06 |
| DFPN-E | 100 | 15 | 8.31 | 23 | 17 | 6.00 |

The metrics LRN and SRN are the number of longest routes and the number of shortest routes.
The metric Avg is the average length of all routes.
Values marked with bold are the best performance under each metric.

We illustrate two solution routes for the same target molecule (CAS NO.:1374357-00-2) given by our EG-MCTS and Retro*+ in Fig. 3. The dashed box part shows the differences between EG-MCTS and Retro*+. In terms of route length, our approach leaves out one extra step in the middle. Note that in this paper we do not consider other chemical criteria, such as the final yield, which can be further investigated in the future, as mentioned in the conclusion section.

To evaluate the sensitivity of our collected experience to the chosen search method, we adapted DFPN by replacing the proof number initialization with value $Q$ predicted by our EGN, called DFPN-E+. Similar to previous evaluation, we set the iteration limit as 500. The experimental results are shown in Table 3. We can see that with the guidance of our EGN, DFPN-E+ achieves better results in both planning efficiency and route quality, compared to the original DFPN-E. This suggests that our learned experience can assist different search methods for retrosynthetic planning and achieve performance improvements rather than heavily biasing search methods used to gather experience.

**Experiments on transferability**. We would like to investigate the transferability of our EGN model on different datasets. To do this, we use ChEMBL as another set of building blocks, which consists of 2.3$M$ bioactive molecules with drug-like properties. After extraction, we obtain 2396 molecules for training, 305 for validation, and 296 for testing. The detailed extraction process is in the section "Datasets". Furthermore, to see the change of performance with respect to different sizes of training sets, we extract two new training sets on ChEMBL using the same sampling procedure. These two new training sets, denoted as $\mathbb{T}_1$ and $\mathbb{T}_2$, having 2500 compounds, respectively, while the initial training set, denoted as $\mathbb{T}_0$. In addition, we merge $\mathbb{T}_0$ with $\mathbb{T}_1$ to obtain $\mathbb{T}_{0+1}$ and $\mathbb{T}_0$ with $\mathbb{T}_1$ and $\mathbb{T}_2$ to obtain $\mathbb{T}_{0+1+2}$ to explore the effect of training set size on EGN.

We compared EG-MCTS, Retro*+ to their non-learning versions, EG-MCTS-0 and Retro*-0+. The experimental results are shown in Table 4. Note that since the value network used in Retro*+ is trained based on eMolecules, we used the same process as described in the paper[7] to extract synthetic routes from ChEMBL and trained the value network based on the extracted routes, denoted by Retro*+(ChEMBL). Similarly, we use EG-MCTS(ChEMBL) and EG-MCTS(eMol) to denote the value network of EG-MCTS is trained from ChEMBL and eMolecules, respectively. And we use Retro*+(eMol) to denote the value network of Retro*+ is trained from eMolecules. The experimental results are all conducted with 500 iterations.

In the first part of Table 4, we show the experimental results on the test set of 296 molecules from ChEMBL via transferring the value network learnt from eMolecules. We can see that EG-MCTS(eMol), which directly transfers the EGN network learnt from eMolecules to synthesize routes for target molecules from ChEMBL, outperforms all of the other approaches, which verifies our proposed framework has better transferability, compared to approaches without transferred value networks (i.e., EG-MCTS-0 and Retro*-0+) and Retro*+(eMol) that transfers the value network learnt from eMolecules to synthesize routes for target molecules from ChEMBL. We can also see that the "success rate" and "Avg iter" of EG-MCTS(ChEMBL), marked with underline, are better than EG-MCTS(eMol). This is consistent with our intuition since retraining the value network from ChEMBL for synthesizing routes for target molecules from ChEMBL should be better than directly transferring the value network learnt from eMolecules, provided that there are sufficient training data from target ChEMBL. Note that we aim to evaluate the transferability

**Fig. 3 Solutions given by EG-MCTS and Retro\*+ for the same target (CAS NO.:1374357-00-2).** The dashed box part shows the differences between EG-MCTS and Retro\*+. Retro\*+ requires an extra step. The molecules over the arrow are from $\mathcal{B}$. The atoms and bonds marked red are reaction center, which change in the reaction.

**Table 3 The performance of EG-MCTS and DFPN-E+ on our test set of 180 molecules and Retro\*-190.**

| Algorithm | Planning efficiency | | Route quality |
|---|---|---|---|
| | Success rate | Avg iter | Avg len |
| Performance on our test set of 180 molecules | | | |
| EG-MCTS-0 | 81.11 | 128.96 | 8.15 |
| EG-MCTS | **94.44** | **60.75** | 5.85 |
| DFPN-E | 76.67 | 170.34 | 8.31 |
| DFPN-E+ | 85.00 | 116.59 | **5.75** |
| Performance on Retro\*-190 | | | |
| EG-MCTS-0 | 73.68 | 186.15 | 5.87 |
| EG-MCTS | **96.84** | **55.84** | 5.07 |
| DFPN-E | 75.26 | 208.12 | 6.00 |
| DFPN-E+ | 85.26 | 128.77 | **4.34** |

The metric Avg iter is the average number of iterations. The metric Avg len is the average of length of all routes.
Values marked with bold are the best performance under each metric.

of our proposed framework under the condition that there are no training data for learning the value network from the target ChEMBL. It is not our focus to compare to the case that there are sufficient training data to learn the value network from the target ChEMBL as done by EG-MCTS(ChEMBL), since transferability may not be needed when there are sufficient training data for learning the network of high-quality.

We can also see that Retro\*+(ChEMBL) and Retro\*+(eMol) perform similarly which indicates that Retro\*+ is insensitive to the value network. The success rate of the two approaches is less than 50% and the retrained network on ChEMBL (i.e., Retro\*+(ChEMBL)) brings performance regressions compared to Retro\*+(eMol), which is inconsistent with our intuition, i.e., the value network learnt from the target ChEMBL is supposed to be better than directly using the value network learnt from the source eMolecules. We conjecture that the way of extracting training routes (as done by Retro\*+(eMol)) for learning Retro\*+(ChEMBL)) is not applicable to small datasets like ChEMBL. Specifically, training set extracted from ChEMBL only has 93,369 items after data equalization process, which is far <299,202 items extracted from eMolecules.

We also would like to see the transferability of our approach from ChEMBL to eMolecules. We directly use the value network learnt from ChEMBL as the one to synthesize routes for the molecules in our test set and Retro\*-190 with eMolecules as the set of building blocks. The results are shown in the second part and the third part of Table 4, respectively. We can see that EG-MCTS(ChEMBL) performs better than EG-MCTS-0, which means our EGN is able to learn knowledge that are transferable from ChEMBL to synthesize routes for target molecules from eMolecules. Likewise, we aim to evaluate the transferability of our

proposed framework under the condition that there are no training data for learning the value network from target eMolecules. It is not our focus to compare to the case that there are sufficient training data to learn the value network from the target eMolecules as done by EG-MCTS(eMol) (marked with underline in Table 4), since transferability may not be needed when there are sufficient training data for learning the network of high-quality.

Table 5 shows the performance of EG-MCTS with EGN trained from different training sets extracted from ChEMBL. Although the performance of EG-MCTS varies with respect to different training sets, i.e., $\mathbb{T}_0$, $\mathbb{T}_1$, and $\mathbb{T}_2$, there is obvious improvement brought by EGN over EG-MCTS-0. Comparing the performance of EGN trained on $\mathbb{T}_0$, $\mathbb{T}_{0+1}$, and $\mathbb{T}_{0+1+2}$, whose size is increasing in order, we can see that the increase of the size of training set does not necessarily improve the performance. This indicates that it would be more helpful to find representative molecules with synthetic experience to constitute a modest training set, instead of using all molecules as the training set without principle. In other words, combining all training sets altogether may not be helpful for improving the performance.

**EG-MCT versus literature.** In order to verify the validity of the routes our EG-MCTS generated, we compare the routes generated by EG-MCTS as well as others with the published routes for 30 testing molecules. The information of 30 testing molecules refer to Supplementary Table 1. Similar to previous work[7,10], we set the maximal number of iterations to be 500 for each target molecule. The difference is that we will not stop the search until 500 iterations have been run out, so for each target molecule, multiple routes can be found. We only choose the route that best matches the published route. Then we calculate the matching degree between the best route and the published route for each test molecule. The calculation procedure of the matching degree is that if the step of the route appears in the published route, and is in the same order as the published route, it is considered that the step is matched. Note that we only match the decomposition reactants and the main products, and do not care about the by-products. We use the number of matching steps divided by the number of steps of generated route as the matching degree. Figure 4 shows the statistics of the matching degree over 30 test molecules. 66.67% of the routes EG-MCTS generate match more than 80%. MCTS-rollout is the second-best performing approach, with 60% of molecules having a match rate ≥80%. Retro\*+ and DFPN-E perform relatively similarly, slightly behind MCTS-rollout.

There are 40% of the generated routes by EG-MCTS that almost exactly match the published routes. Note that "almost exactly match" indicates the each step of generated routes appears in the published routes but the final molecules (buliding blocks) in the generated routes continue to be decomposed in the

**Table 4 The results of experiments on transferability.**

| Algorithm | Planning efficiency | | Route quality |
|---|---|---|---|
| | Success rate | Avg iter | Avg len |
| Performance on the test set of ChEMBL with ChEMBL as $\mathcal{B}$ | | | |
| EG-MCTS-0 | 59.12 | 278.44 | 11.56 |
| EG-MCTS(eMol) | **62.16** | **272.08** | **8.36** |
| EG-MCTS(ChEMBL) | 79.05 | 164.28 | 9.77 |
| Retro*-0+ | 41.22 | 355.77 | 10.43 |
| Retro*+(eMol) | 48.65 | 327.67 | 10.22 |
| Retro*+(ChEMBL) | 47.97 | 332.03 | 10.44 |
| Performance on our test set of 180 molecules with eMolecules as $\mathcal{B}$ | | | |
| EG-MCTS-0 | 81.11 | 128.96 | 8.15 |
| EG-MCTS(ChEMBL) | **93.33** | **79.96** | **7.71** |
| EG-MCTS(eMol) | 94.44 | 60.75 | 5.85 |
| Performance on Retro*-190 with eMolecules as $\mathcal{B}$ | | | |
| EG-MCTS-0 | 73.68 | 186.15 | **5.87** |
| EG-MCTS(ChEMBL) | **84.21** | **123.27** | 6.63 |
| EG-MCTS(eMol) | 96.84 | 55.84 | 5.07 |

EG-MCTS(eMol) and Retro*+(eMol) use the EGN trained on eMolecules, while EG-MCTS(ChEMBL) and Retro*+(ChEMBL) use the EGN trained on ChEMBL. The metric Avg iter is the average number of iterations. The metric Avg len is the average of length of all routes.
Values marked with bold are the best performance under each metric.
Since we aim to evaluate the transferability, those retraining approaches, marked with underline, are not involved in the comparison.

**Table 5 The performance of EG-MCTS with respect to different sizes of training sets on the test set of ChEMBL using ChEMBL as $\mathcal{B}$ under the iteration limit of 500.**

| Algorithm | Planning efficiency | | Route quality |
|---|---|---|---|
| | Success rate | Avg iter | Avg len |
| EG-MCTS-0 | 59.12 | 278.44 | 11.56 |
| EGN on $\mathbb{T}_0$ | 79.05 | 164.28 | 9.77 |
| EGN on $\mathbb{T}_1$ | 75.00 | 200.14 | 9.55 |
| EGN on $\mathbb{T}_2$ | 74.32 | 208.38 | **9.39** |
| EGN on $\mathbb{T}_{0+1}$ | **80.74** | 164.30 | 10.08 |
| EGN on $\mathbb{T}_{0+1+2}$ | 80.41 | **156.78** | 10.11 |

The metric Avg iter is the average number of iterations. The metric Avg len is the average of length of all routes.
Values marked with bold are the best performance under each metric.

published routes. Figure 5 shows an exemplary 11-step route generated by our EG-MCTS for the molecule (CAS NO.:1392842-01-1) of inhibiting HIF hydroxylase enzyme activity reported in 2012, which fully matches the published route in the patent[30]. The other three approaches choose other way in the last step as shown in Fig. 5b.

Another 40% of the generated routes by EG-MCTS mostly match the published routes, with an average matching ratio of 77.23%. We observe that the difference mainly occurs in the later part of the retrosynthetic routes, while the routes are completely, especially in the first 5 to 7 steps. We also observed that there are two main differences. One is that because our EG-MCTS is goal-oriented, i.e., to break target molecules into building blocks, EG-MCTS gives priority to the successful decomposition ways which are different from the published routes, as the step 8 in Fig. 6a compared to steps 8 to 10 in Fig. 6b. Note that identical steps 1 to 7 are not shown in the Fig. 6. The route shown in Fig. 6b is reported in the patent[31]. In step 8 of EG-MCTS in Fig. 6a, compound **a.2**, methyl 4-chloro-2, 3-diaminobenzoate is reacted with 3-Bromopropyl alcohol. But it may not work since EG-MCTS chooses the less reactive amino group between the two amino groups in compound **a.2**. We also observe that the other three approaches generate the same routes as ours. Another is that although the intermediate decomposition steps are different, the final decomposition results are identical. As shown in Fig. 7, the generated route given by EG-MCTS and the published route reported in the patent[32] have different intermediate steps, i.e., steps 8–11 in Fig. 7a and steps 8–10 in Fig. 7b, but have the same intermediate decomposition compound **b.12**, which is in the red dotted frame. In the generated route, the carboxylic ester (**b.12**) is firstly reduced to the alcohol (**b.11**) in step 11, and in step 10 the alkyl halide (**b.10**) is obtained from the alcohol (**b.11**) by chlorination. These two reactions have been included in the patent[33]. Step 9 is the substitution reaction of the alkyl halide (**b.10**) with cyanide reagent and produces the nitrile-containing compound **b.9**. Step 8 is the alcoholysis of nitriles to esters under the catalysis of acids. The number of steps of the generated route is one more than the published route, but each step also seems to be acceptable. The other three approaches end early by choosing other decomposition way at Step 7 as shown in Fig. 7c.

There are 6 of 30 generated routes by EG-MCTS whose matching degree is lower than 60%. Figure 8 shows a route different from the published route reported in the patent[34]. Step 10 is the acylation of acid chloride and the amine (**c.11**) to the amide (**c.10**) and step 9 is the substitution reaction of alcohol hydroxyl of compound **c.10** with trifluoromethanesulfonic anhydride to provide the trifluoromethanesulfonyl of compound **c.9**. In step 8, the amide group of compound **c.9** undergoes the amidohydrolysis. Step 7 is the substitution reaction that turns the secondary amine (**c.8**) to the tertiary amine (**c.7**). Step 6 is the coupling of aryl compounds with arylboronic acid derivatives (Suzuki Coupling) and step 5 is the halogenation of aromatic compounds, both of which have been included in the patent. The substitution reaction on alkyl halide (**c.5**) with cyanide reagent gives the nitrile-containing compound **c.4** in step 4. Compound **c.4** is then deprotected to the lactam by demethylation in step 3. The ester group of compound **c.3** is then hydrolyzed to the acid in step 2. In the last step, compound **c.2** is aminated to give the amide (**c.1**).

Although each step of these routes follows some chemical reaction principles, some intermediate molecules of these routes may not exist in reality or have not yet been synthesized, due to the failure to consider the chemical environment. For example, the groups of the molecule itself cannot coexist and the positions and groups at which reactions can occur are various and do not definitely proceed as they do in the planning routes. After searching, we could not find the CAS number of compounds **c.2, c.3, c.4, c.8, c.9, c.10** appearing in the route shown in Fig. 8, which means that they may not exist in reality or have not yet been synthesized. These disturbing problems are common in existing retrosynthetic planning approaches.

**Drug retrosynthetic planning.** We apply our EG-MCTS approach to the synthesis of some commercialized star drug molecules with complex structures to find out whether the planning synthetic routes have practical guiding significance. Here are five used molecules in the drug retrosynthetic planning experiments: mannopeptimycin aglycone, Paxlovid, Sofosbuvir, Taxol, and Molnupiravir.

The first drug molecule is mannopeptimycin aglycone, which is the cyclic hexapeptide aglycone of the mannopeptimycins, a group of glycopeptides known for potent activity against drug-resistant bacteria. The CAS number of mannopeptimycin aglycone is 1622135-35-6. We ignore its stereochemical structure to get the target molecule for EG-MCTS, as compound **d.1** shown in Fig. 9a. The generated retrosynthetic route for mannopepti-mycin aglycone is shown in Fig. 9a. Note that template-based approaches (including EG-MCTS) sometimes ignore some side

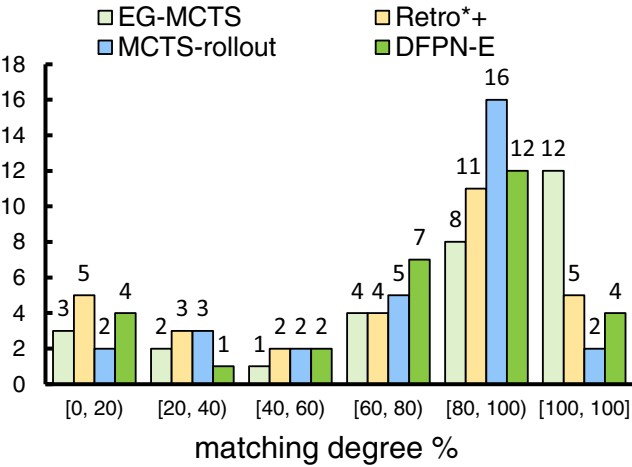

**Fig. 4 The statistics of the matching degree.** We compare the routes generated by EG-MCTS as well as other three approaches with the published routes for 30 testing molecules.

intermediates, which may incur confusion for readers to understand the generated routes. To help readers understand the routes, we randomly specified one intermediate that can participate in the reaction and marked it in blue.

The designed route starts from the esterification of compound **d.8** with ethanol (step 7). In step 6, the amine group of compound **d.7** undergoes the condensation reaction with the carboxyl group of aid (the blue compound) to form the amid (**d.6**). Step 5 is the hydrolysis of the ester into the carboxyl. Compound **d.5** then undergoes the acylation reaction with Methyl 2-amino-3-(4-hydroxyphenyl)propanoate in step 4, followed by two successive condensation reactions of carboxyl and amino groups in steps 3 and 2. The last step is the intramolecular acylation reaction, in which the amine group and the ester group of compound **d.2** are involved to form a hexapeptide ring.

The second molecule is Paxlovid, which is the first oral antiviral drug authorized by the FDA for the treatment of COVID-19. The CAS number of Paxlovid is 2628280-40-8. We also ignore its stereochemical structure and use our EG-MCTS to get its retrosynthetic route as shown in Figure. Note that we also add the necessary intermediates to the route and mark them in blue.

The generated route starts with two building blocks, compound **e.7** and **e.10**. The hydroxyl group of compound **e.10** undergoes the esterification reaction with oxalyl chloride in step 8 and the amide (**e.9**) is then hydrolyzed in step 7. On the other side, compound **e.7** first reacts with bromoacetic acid to produce the acid derivative (**e.6**) in step 6 and then the carboxylic acid (**e.6**) is reduced to the aldehyde (**e.5**) in step 5. In step 4, compounds **e.5, e.8** and cyanide participate in a ternary reaction to get the compound **e.4** with the cyano group and the amide group. Step 3 is the amide hydrolysis and generates compound **e.3**. In step 2, a condensation reaction occurs between compound **e.3** and the compound, which is above the "step 2" arrow, and the amide hydrolysis occurs at the same time, resulting in compound **e.2**. The last step is also the condensation reaction of carboxyl group and amino group, happening between compound **e.2** and trifluoroacetic acid.

The generated routes for the other three drugs are listed in Supplementary Fig. 1. From the two routes, even ignoring the stereochemical structure and some reactants, our generated routes are definitely not perfect. There are many details to be perfected, such as whether the presence of intermediate compounds is reasonable and whether the reactions will go as

planned. For a specific example, the structural stability of compound **e.8** in the Fig. 9b is questionable, as acylation may occur between the amine group and the acid chloride inside compound **e.8**. Although the generated routes given by our EG-MCTS are not mature enough, but they are heuristic for synthetic organic chemists while performing retrosynthesis for complex compounds and can guide them in which direction to consider. It would be even more helpful if chemists could adjust the generated routes according to these imperfect and inaccurate details and finally get a relatively feasible path. For example, for the detail of the structural instability of compound **e.8**, we can make minor adjustments to the generated route as shown in Fig. 9c. In the adjusted route, we use compound **e.9** instead of compound **e.8** to participate in the ternary reaction with compound **e.5** and cyanide to generate new compound **e.11** (step 3). The two amide groups of compound **e.11** are then hydrolyzed at the same time in step 3, discarding the two tert-butyl hydrogen carbonate. Small adjustments like this make the resulting routes more reasonable.

## Conclusion

In this paper, we propose EG-MCTS, a novel MCTS-based retrosynthetic planning approach. Different from existing machine-trained approaches which are limited to the existing datasets, we investigate the way of acquiring chemical synthetic knowledge and experience. Our experimental results on real-world benchmark datasets exhibit our EG-MCTS gains significant improvement over existing approaches. The comparison between the generated routes and the published routes also confirms the validity and feasibility of our approach. We use our EG-MCTS to perform retrosynthetic planning for realistic drugs or compounds, and the results exhibit that EG-MCTS is instructive. At the same time, the experiments on real compounds have exposed the inadequacies of our approach, which are also common problems of retrosynthetic planning approaches, that is, the understanding and learning of chemical reaction principles are not thorough and comprehensive. It can be embodied as whether the presence of compounds is reasonable and whether the reactions will go as planned and so on. We believe that if these problems are solved, the quality of the generated routes can be greatly improved.

In this work, we did not consider reagents and other chemical reaction conditions, which could be different from actual chemical reactions. In addition, in the input of our EGN, molecules and reaction templates are represented as fixed-dimensional fingerprints, which could incur bit collisions. In the future we will investigate the possibility of exploring machine learning-based approaches to making up for the above-mentioned limitations. Finally, we measure route quality by the length of the route, which is relatively simple and may cause the algorithm to choose a strategy of removing some protection steps, making the routes different from real reactions. In the future we will explore some chemically-meaningful evaluation metrics.

In addition, in planning community, there have been techniques of high-performance with respect to planning and learning Zhuo and Kambhampati[35], Zhuo and Yang[36], Zhuo et al.[37,38], Shen et al.[39]. It would be interesting to investigate "borrowing" those techniques to deal with the retrosynthetic planning problem in the future.

## Methods

We first describe the RS planning problem as a Markov Decision Process. Then we introduce the key part, EG-MCTS planning and the two phases of EG-MCTS approach in detail. Finally, we introduce the datasets and baseline approaches.

**Retrosynthetic planning problem**. RS planning can be viewed as a Markov Decision Process (MDP)[40], defined by a state space $\mathcal{S}$, an action space $\mathcal{A}(s)$, a

**Fig. 5 An exemplary 11-step route generated for the molecule (CAS NO.:1392842-01-1) by EG-MCTS which matches the published route. a** Chemical solution route given by EG-MCTS. **b** Chemical solution route given by other three approaches. The molecules over the arrow are from $\mathcal{B}$. The atoms and bonds marked red are reaction center, which change in the reaction. In **b**, the route from the target to the molecule in the dashed box is consistent with ours.

**Fig. 6 A highly-matching example showing the differences between EG-MCTS generated route and the published one[31]. a** Chemical solution route given by EG-MCTS, Retro*+, DFPN-E and MCTS-rollout. **b** Published solution route. Steps 1 to 7 are the same and not shown in the figure. The intermediate molecule **a.1** are then decomposed in two different ways. The CAS Number of the target molecule is 1173980-10-3. The molecules over the arrow are from $\mathcal{B}$. The atoms and bonds marked red are reaction center, which change in the reaction.

transition model $\mathcal{T}(s, a, s')$, a policy $\pi(a|s)$ and a reward function $\mathcal{R}(s, a, s')$. In RS planning, a state $s \in \mathcal{S}$ is a set of molecules, and the initial state $s_0 = m_0$ is composed of the target molecule $m_0$. Actions are reaction templates applied to one of the molecules $m$ in state $s$. The transition function $\mathcal{T}(s, a, s')$ is deterministic for simplicity. The policy $\pi(a|s)$ is the probability distribution of all allowed functions. The reward function $\mathcal{R}(s, a, s')$ can be simplified as $\mathcal{R}(m, T)$, indicating the reward taken by applying reaction template $T$ on molecule $m$.

**EG-MCTS planning**. We first introduce the key part, EG-MCTS Planning. We observe that AND-OR tree structure is suitable for RS planning[6,7,10,41], capturing the relations between reactions and corresponding molecules.The result of EG-MCTS planning can be represented as an AND-OR tree.

An AND-OR tree has two different types of nodes, i.e., AND node that succeeds only if all of its child nodes are successful, and OR node that succeeds only if at least one child node is successful. In RS planning, a molecule is viewed as successful if there exists at least one reaction that can break it down to $\mathcal{B}$. A reaction is viewed as successful if all of its reactants are successful. The retrosynthetic searching process can be represented as an AND-OR tree, whose OR and AND nodes are molecules and reaction templates, respectively. Note that a reaction template can be seen as a reaction relation among substructures of reactants and products. For example, "$\bar{x} \rightarrow \bar{a} + \bar{b}$" is a reaction template, where $\bar{x}$, $\bar{a}$, $\bar{b}$ are substructures of molecules $x$, $a$ and $b$ in reaction "$x \rightarrow a + b$", respectively. In EG-MCTS planning, the OR node (molecule node) contains molecule and a value $V_m$, and the AND node (reaction node) contains a reaction template and a value $\bar{Q}$. We denote a molecule node $m$ as successful if its

molecule belongs to $\mathcal{B}$ or one of its child reaction nodes is denoted as successful. We denote a molecule node as unsuccessful if all of its child nodes are denoted as unsuccessful or there is no reaction template available to be applied to $m$. Likewise, we denote a reaction node $T$ as successful if all of its child molecule nodes are denoted as successful, and denote it as unsuccessful if one of its child nodes is denoted as unsuccessful.

We address the three modules of EG-MCST planning in detail below.

- Selection: in order to select a promising molecule node, we need to build a selection module to repeatedly select reaction templates for molecule nodes and (sub-)molecules for reaction nodes, until a leaf molecule node is found. Intuitively, for a molecule node, we select the most promising reaction templates based on the PUCT policy as used by[29], as shown in Eq. (1):

$$T^* = \arg\max_{T \in child(m)} \left( \frac{\bar{Q}(m, T)}{N(T)} + cP(m, T) \frac{\sqrt{N(T')}}{1 + N(T)} \right) \quad (1)$$

In Eq. (1), $\bar{Q}(m, T)$ is an average score over all previous scores, which will be repeatedly updated according Eq. (3) given by the Update module. $P(m, T)$ is given by the single-step retrosynthetic model $S(\cdot)$, and $N(T)$ records the number of times that node $T$ has been updated. $T'$ is the grandparent reaction node of the reaction node $T$. The exploration constant $c$ is a hyper-parameter. For a reaction node, if it has child nodes which have not been expanded, the algorithm will give priority to this kind of child nodes and randomly choose one. Otherwise, randomly select one among the child nodes which have not been proved successful.

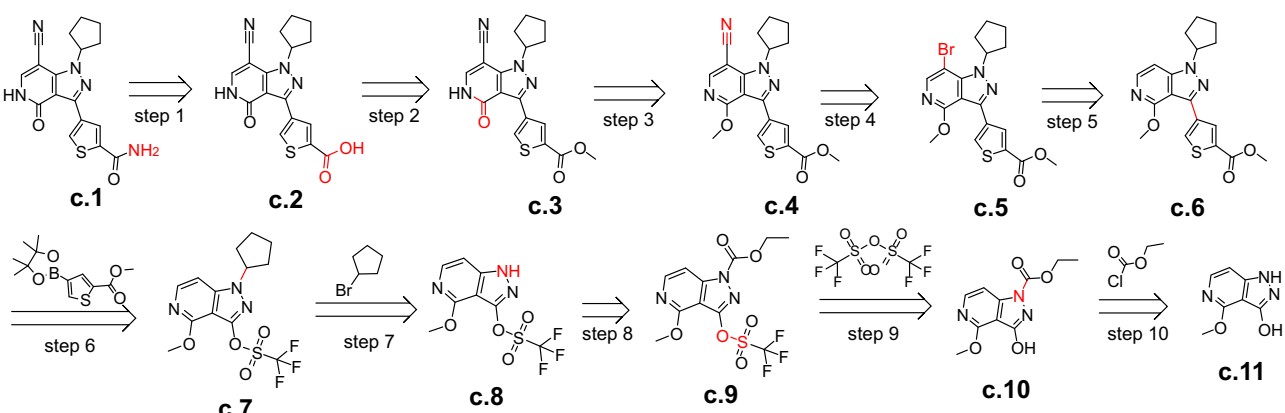

**Fig. 7 Another highly-matching example showing the differences between EG-MCTS generated route and the published one[32]. a** Chemical solution route given by EG-MCTS. **b** Published solution route. **c** Chemical solution route given by Retro*+, DFPN-E and MCTS-rollout. **a**, **b** share the same Steps 1 to 7. **b**, **c** share the same Steps 1 to 6. The CAS Number of the target moleuclue is 1443043-01-3. The molecules in the dashed box are the same. The molecules over the arrow are from $\mathcal{B}$. The atoms and bonds marked red are reaction center, which change in the reaction.

**Fig. 8 A lowly-matching example route given by EG-MCTS.** The molecules over the arrow are from $\mathcal{B}$. The atoms and bonds marked red are reaction center, which change in the reaction.

- Expansion: The single-step retrosynthetic model $S(\cdot)$ is applied to the molecule $m$ contained in the selected molecule node, and it predicts the top-$k$ promising reaction templates. If the output set is empty, indicating no available reaction templates, the node is unsuccessful. Otherwise, each reaction template $T_j$ is added to the tree as a child reaction node of the selected molecule node with $\bar{Q}(m, T_j) = Q_0(m, T_j)$ given by the EGN. After

applying the template $T_j$ on $m$, we get the corresponding reactant set $R_j$. Each reactant $r$ in $R_j$ is also added as a child molecule node of the reaction node $T_j$.

- Update: The update step starts from the selected molecule node and upwards along the tree. At the molecule node, the algorithm checks whether the node is successful or unsuccessful. If it is not proved to be

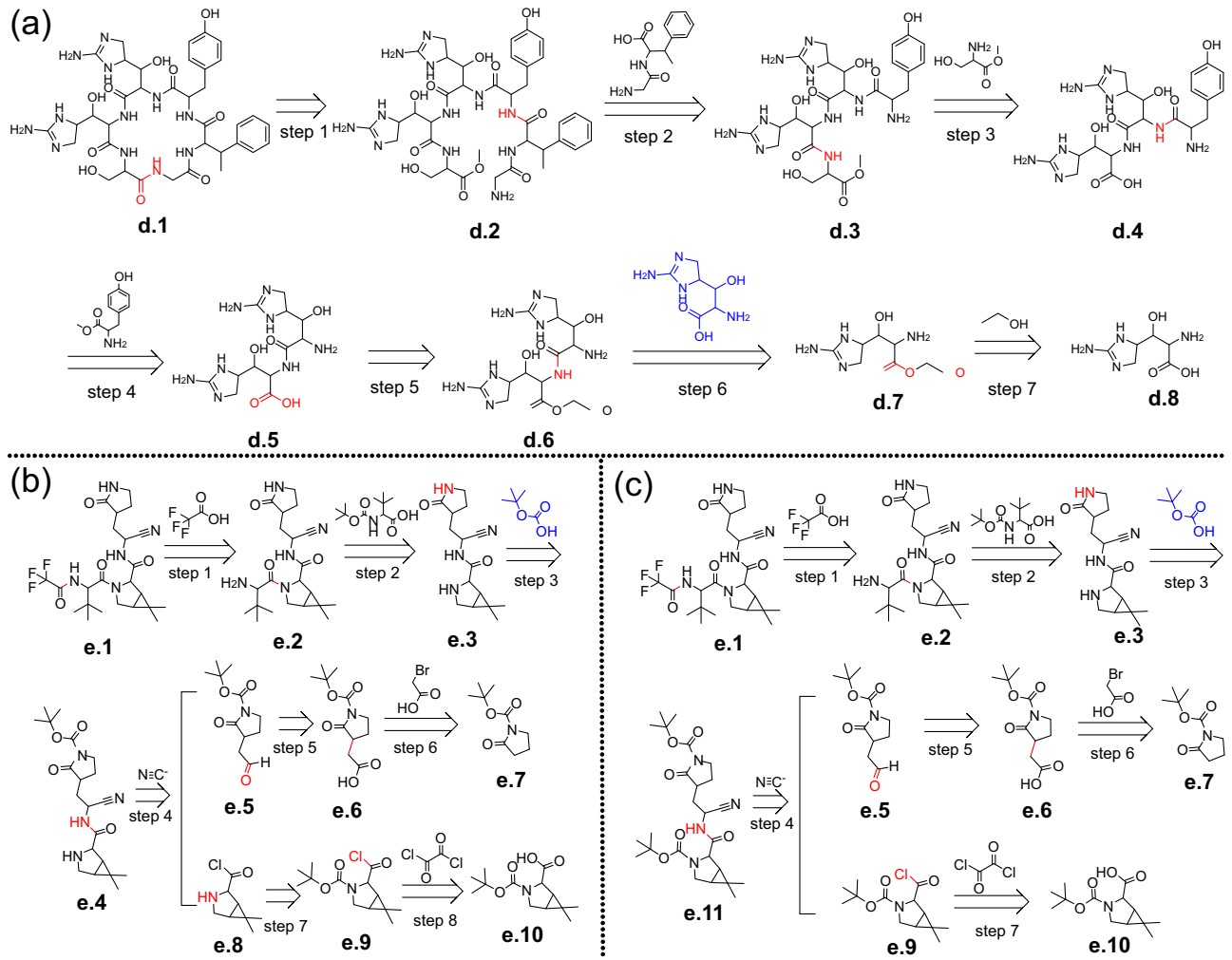

**Fig. 9 Two example in drug retrosynthetic planning experiment. a** The generated route given by EG-MCTS for mannopeptimycin aglycone, whose CAS number is 1622135-35-6. **b** The generated route given by EG-MCTS for Paxlovid, whose CAS number is 2628280-40-8. **c** The adjusted route according to **b** for mannopeptimycin aglycone. In experiment, we ignore their stereochemical structure. The molecules over the arrow are from $\mathcal{B}$. The atoms and bonds marked red are reaction center, which change in the reaction. We add the necessary intermediates to the route and mark them in blue.

unsuccessful, the algorithm updates its $V_m$ to the highest $\bar{Q}$ among its child nodes:

$$V_m(m) = \max_{T \in child(m)} \bar{Q}(m, T) \quad (2)$$

At the reaction node, the algorithm firstly updates its update count $N(T) = N(T) + 1$. Then the algorithm records its $Q$ value in the $N(T)^{th}$ update, denoted as $Q_{N(T)}(m, T)$. $Q_{N(T)}(m, T)$ is given by the reward function $\mathcal{R}(m, T)$. The reward function returns $z > 1$ if the reaction node is proved to be successful, and $-z$ if it is unsuccessful. Otherwise, the reward function calculates the average $V_m$ among its child nodes. After getting the reward in the $N(T)^{th}$ update, the algorithm updates the average score $\bar{Q}$ of the reaction node:

$$\bar{Q}(m, T) = \frac{1}{N(T) + 1} \sum_{j=0}^{N(T)} Q_j(m, T) \quad (3)$$

Note that $Q_0(m, T)$ is given by EGN when the reaction node $T$ is added to the tree, which is not counted in its update count, and $Q_j(m, T), j \in [1, N(T)]$ is given by the reward function.

**Phase I: learning EGN.** The detailed learning procedure can be found from the algorithm shown Fig. 10. We first initialize the EGN with random weights $\theta_0$, which is denoted by $f_{\theta_0}$. At each training round $i \geq 1$, for each target molecule $m \in \mathcal{M}_{train}$, we build a search tree $\mathcal{T}_m$ using EG-MCTS planning with $f_{\theta_{i-1}}$ (Step 5). We then collect the training data based on $\mathcal{T}_m$ (Step 6). After that we update the EGN with the training data and get the new EGN $f_{\theta_i}$ (Step 9). We verify the

performance of the new EGN on the validation molecule set, i.e., perform EG-MCTS-planning for each molecule $m \in \mathcal{M}_{validation}$ (Step 11). If the success rate and average number of iterations can not satisfy the loop condition $\mathcal{L}_{loop}$, then the learning algorithm stops and return the well-trained EGN. In the following sub-sections, we will address three procedures experience collecting, EGN updating and EGN validating of the algorithm, respectively.

- Experience collecting: The Experience Guidance Network learns from chemical synthetic experience and uses experience to guide the future search. It takes a reaction template $T$ and a molecule $m$ as inputs, then predicts the score of template $T$ acting on molecule $m$. It works based on the following assumptions:

1. The score of a reaction template acting on a molecule is independent of others, so independent prediction is reasonable.
2. The same decomposition action $(m, T)$ may appear in the search of different target molecules, so EGN, which has learned the value of action $(m, T)$ from past searching, will give a more accurate value while meeting the same action.
3. The most potential reaction templates of two similar compounds are likely to be the same. The well-trained network which has learned from the past synthetic experience showing that the reaction template $T$ works well in molecule $m$ will encourage the search to select $T$ when similar molecule $m'$ is encountered.

Specifically, in the $i^{th}$ round of training of the EGN, for every molecule $m$ in the training set $\mathcal{M}_{train}$, EG-MCTS planning builds a search tree $\mathcal{T}_m$. For every reaction node $T$ in the tree $\mathcal{T}_m$, it and its parent molecule node $m$ composes a decomposition action $(m, T)$. We collect every decomposition action $(m, T)$ and the $\bar{Q}$ stored in the corresponding reaction node $T$ to form

---

**Algorithm 1: Learning EGN**

---

**Input:** Training molecule set $\mathcal{M}_{train}$, validation molecule set $\mathcal{M}_{validation}$, building blocks set $\mathcal{B}$, one-step retrosynthetic model $S(\cdot)$

**Output:** well-trained EGN $f_\theta$

1: Initialize EGN $f_{\theta_0}$ with random parameters $\theta_0$;
2: **for** i=1, $max\_round$ **do**
3: Initialize a training data $\mathcal{D}^i_{train} = \{\}$;
4: **for** $m \in \mathcal{M}_{train}$ **do**
5: Build search tree $\mathcal{T}_m$:
 $\mathcal{T}_m = EG\text{-}MCTS\text{-}planning(m, \mathcal{B}, S(\cdot), f_{\theta_{i-1}})$;
6: Collect training data $\mathcal{D}$ from $\mathcal{T}_m$:
 $\mathcal{D} = experience\text{-}collecting(\mathcal{T}_m)$;
7: $\mathcal{D}^i_{train} = \mathcal{D}^i_{train} \cup \mathcal{D}$;
8: **end for**
9: Update EGN with $\mathcal{D}^i_{train}$:
 $f_{\theta_i} = EGN\text{-}updating(f_{\theta_{i-1}}, \mathcal{D}^i_{train})$;
10: **for** $m \in \mathcal{M}_{validation}$ **do**
11: search $m$ using EG-MCTS-planning:
 $EG\text{-}MCTS\text{-}planning(m, \mathcal{B}, S(\cdot), f_{\theta_i})$;
12: **end for**
13: Complete the success rate $\mathcal{R}_{s_i}$ and the average number of iterations $\mathcal{R}_{a_i}$ on $\mathcal{M}_{validation}$;
14: Complete the highest success rate $\mathcal{R}_{s_{max}}$ and the lowest average number of iterations $\mathcal{R}_{a_{min}}$ of the last five rounds on $\mathcal{M}_{validation}$;
15: **if** $\mathcal{L}_{loop}(\mathcal{R}_{s_i}, \mathcal{R}_{s_{max}}, \mathcal{R}_{a_i}, \mathcal{R}_{a_{min}})$ is false **then**
16: **return** $f_{\theta_i}$;
17: **end if**
18: **end for**

---

**Fig. 10 The algorithm of Phase I Learning EGN.** The EGN is first randomly initialized, and then experience collecting, EGN updating, and EGN validating are performed sequentially in each iteration.

the experience set $\mathcal{D}^i_{train} = \{(m, T), \bar{Q}\}$. If the same decomposition action $(m, T)$ occurs multiple times in the experience set, we unify their Q values to the mean of the scores according to the multiple occurrences.

- EGN updating: the EGN is a single-layer fully connected neural network with input dimension of 4096 and hidden dimension of 256. It outputs a scalar $Q \in (0, 1)$ representing the predicted value. At training round $i$, the neural network $Q = f_{\theta_{i-1}}(m, T)$ is trained for 20 epochs on dataset $\mathcal{D}^i_{train}$ to minimize $\mathcal{L}_{MSE}$, using Adam optimizer[42]. We apply dropout[43] as a means of regularization with the dropout rate 0.1.

$$\mathcal{L}_{MSE} = \left(Q - \bar{Q}(m, T)\right)^2 \tag{4}$$

- EGN validating: we then verify the new EGN $f_{\theta_i}$ on the validation set. Specifically, the algorithm records the highest success rate $\mathcal{R}_{s_{max}}$ and the lowest average number of iterations $\mathcal{R}_{a_{min}}$ of the last five training round on the validation set. At the training round $i$, the algorithm completes the success rate $\mathcal{R}_{s_i}$ and the average number of iterations $\mathcal{R}_{a_i}$ of EG-MCTS planning with $f_{\theta_i}$ on the validation set. The loop condition $\mathcal{L}_{loop}$ can be expressed as: $\mathcal{L}_{loop}$ is true if $\mathcal{R}_{s_i} - \mathcal{R}_{s_{max}} > \varepsilon_1$ or $\mathcal{R}_{a_{min}} - \mathcal{R}_{a_i} > \varepsilon_2$. Otherwise, it is false. $\varepsilon_1$ and $\varepsilon_2$ are hyper-parameters.

**Phase II: generating synthetic routes for new target molecules.** To generate synthetic routes for the target molecule $m_0$, we first exploit the EG-MCTS-planning procedure, i.e., Step 5 of the algorithm shown Fig. 10, to generate a tree with the learnt EGN $f_\theta$:

$$EG-MCTS-planning(m_0, \mathcal{B}, S(\cdot), f_\theta).$$

We then initialize a queue with the root node of the tree and an empty reaction list. The following process is repeated until the queue is empty:

- We get the first node $m$ from the queue.
- If $m$ is not from $\mathcal{B}$ and it has a successful child reaction node $T$, we put all children $\{r_j\}^n_{j=1}$ of this reaction node $T$ into the queue and add the reaction

$m \rightarrow \{r_j\}^n_{j=1}$ to the reaction list. If it is not from $\mathcal{B}$ and it does not have a successful child reaction node, the search fails and the reaction list is set to empty.
- If the queue is empty, the search succeeds and the algorithm returns the reaction list.

With the above process, we have the reaction list as the synthetic route of a target molecule.

Note that in our experiment, we empirically set the exploration constant $c$ to be 0.5, the reward $z$ to be 10 for a successful reaction node and $-10$ for a failed reaction node, respectively. We set $\varepsilon_1$ of the loop condition $\Theta$ to be 0.015, and $\varepsilon_2$ to be 3, respectively.

**Datasets.** In order to train the single-step retrosynthetic model $S(\cdot)$, we use the publicly available reaction dataset extracted from United States Patent Office (USPTO) up to September 2016 provided by Lowe[11]. The single-step retrosynthetic model $S(\cdot)$ is a template-based model that treats the template predictzion problem as a multi-class classification problem following previous literature[12,44]. $S(\cdot)$ is trained on the reaction dataset from USPTO with the assistance of RDChiral[14], and the training details refer to literature[7,10]. The input of $S(\cdot)$ is a molecule, and the input of the EGN is the combination of a molecule and a reaction template. We need to represent them by real vectors. For a molecule, we use the Morgan fingerprint of radius 2 with 2048 bits. For a reaction template, its fingerprint could be computed by rdkit, using the function CreateStructuralFingerprintForReaction and the fingerprint is then folded into 2048 dimensions. Note that the function used to calculate the template fingerprint can not encode all chemical information, e.g., number of explicit hydrogens, direct bonds, etc. Therefore, different reaction templates may be represented by the same fingerprint. In practice, there is a degeneration of some fingerprints because of bit collisions since the dimension of fingerprints is fixed, so this issue might not affect the performance of the EGN.

For the experiments in Section 2.3, the building blocks set $\mathcal{B}$ comes from eMolecules, a collection of 231M commercially available molecules. We hope the EGN to have strong generalization ability through learning the synthetic experience of molecules in training set. In order to obtain those molecules with rich and valuable experience, we build a Network of Organic Chemistry (NOC)[45–47] based on USPTO and eMolecules. The NOC is a directed graph, where each node is a molecule and each edge from one node to another, e.g., $A$ to $B$, indicates that there is a reaction where $A$ belongs to its reactants and $B$ to its products.

The *outdegree* of a node $A$ is the number of edges out of $A$. The *cost* of a node $A$ is the length of the longest path among all the paths from $A$ to the leaf nodes in the synthetic tree. We first initialized the directed graph by viewing each molecule in eMolecules as a node in the graph. We then repeated the following procedure until the graph no longer changed:

- We first traversed each reaction in USPTO and added its products to the graph as new nodes if all of its reactants are in the graph.
- For each new node in the graph, we added new edges from each reactant in the graph to the new node.

There are 4650 molecules with *outdegree* $\geq 2$ and *cost* $\geq 4$ in the dataset, among which 907 molecules that are difficult to be solved using Greedy DFS were selected. *Outdegree* $\geq 2$ means the molecule on the synthetic pathways with at least two complex molecules, which is assumed it has richer experience. Since the molecules with higher cost would be broken down to those with lower cost, we collected the experience of those lower-cost molecules during the searching for those with higher cost. We thus put a limit on the cost to avoid experience redundancy. In order to enrich the synthetic experience, we also selected some molecules with higher cost. There are 1499 molecules with *cost* $\geq 9$ in the dataset. To do this, we performed the DFS search and kept the 631 molecules that could not be retro-synthesized successfully within 100 iterations. We then randomly divided the 631 molecules into three subsets: 286, 165, and 180, respectively. The 286 molecules were combined with the 907 molecules mentioned above as the final training set of 1193. The remaining 165 and 180 were used as the validation set and the test set, respectively.

We also use the test set of Retro*[7] and Retro*+[10], called Retro*-190, which consists of 190 molecules. In order to ensure the fairness and effectiveness of the experiment, we do some similarity statistical experiments: for a test molecule $m \in \mathcal{M}_{test}$, we calculate the highest similarity and the average similarity between it and the molecules in the training set, denoted as $S_{max}(m)$ and $S_{avg}(m)$. For all molecules in our test set, the average of $S_{max}$ is 0.62 and the average of $S_{avg}$ is 0.36. And the average of $S_{max}$ in Retro*-190 is 0.61 and the average of $S_{avg}$ is 0.35.

For the experiments in section "Experiments on transferability", the building blocks set $\mathcal{B}$ comes from ChEMBL, which consists of 2.3M bioactive molecules with drug-like properties. Note that ChEMBL contains far fewer molecules than eMolecules. If we use the same method, as done on eMolecules, to extract the training set, validation set and test set, i.e., first constructing the NOC and then filtering the eligible molecules, the number of eligible molecules is small. Considering the difference of sizes between eMolecules and ChEMBL and the scalability of the training set, we use a new method, which different from what we did on eMolecules, as shown below:

1. We first randomly sampled the molecules that are viewed as products in the USPTO reaction dataset.
2. We then randomly initialized the EGN and sought those molecules using EG-MCTS planning with the initialized EGN.
3. We selected the molecules without any successful path being found within 100 iterations.

Since the number of molecules selected with the above-mentioned method is large, about 60,000, we randomly picked 2396 molecules for training, 305 for validation, and 296 for testing, which is almost the same as the number of the training, validation and test sets on eMolecules. $\mathbb{T}_1$ and $\mathbb{T}_2$ use the same sampling procedure as above.

**Baselines**. To verify the effectiveness of EG-MCTS, we compare our approach against other representative baselines in RS planning problem:

1. Retro*+ and Retro*-0+[10] are neural-based A*-like algorithms based on Retro*[7] with a self-improved single-step retrosynthetic model. Retro*+ uses a neural value network trained in the USPTO and Retro-0*+ is its non-learning version. Its code and test set is available.
2. DFPN-E[6] combines the Depth-First Proof-Number (DFPN) Search with Heuristic Edge Initialization. Following the implementation details and parameter settings in the literature, we have implemented DFPN-E.
3. MCTS-rollout uses a basic tree structure whose nodes are molecule sets and edges are reaction templates. The tree structure and search algorithm can be referred to Segler et al.[4]. MCTS-rollout uses rollout to evaluate the values of templates, i.e., algorithm randomly takes a few more steps forward and reaches a future state, then uses the future state score as the current state score. For the max rollout depth, which is the number of forward steps while performing rollout, we used the default maximum depth used by 3N-MCTS, i.e., 5. The exploration constant $c$ is 0.5.
4. Greedy DFS always gives priority to the reaction with the highest probability. We set its max depth to be 10, which is the max depth of the expanded search tree. The node of DFS search tree is defined as a set of molecules, similar to MCTS-rollout.
5. To understand more about the importance of the EGN, we also perform an ablation study by testing the non-learning version EG-MCTS-0 set the initial $Q$ value to be 0.5 for all actions.

All experiments use the same building blocks set $\mathcal{B}$. As for single-step retrosynthetic model $S(\cdot)$, all algorithms use the model of Retro*+[10], except Retro*-0+ (because it has its own model).

## Data availability
All related data in this paper are public. The eMolecules dataset can be downloaded from http://downloads.emolecules.com/free/2019-11-01/. The ChEMBL dataset can be downloaded from https://www.ebi.ac.uk/chembl/. The test set Retro*-190 can be downloaded from https://github.com/binghong-ml/retro_star. The data used in the experiment is available at https://github.com/jjljkjljk/EG-MCTS.

## Code availability
The source code of EG-MCTS is available at https://github.com/jjljkjljk/EG-MCTS.

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

## Acknowledgements

This research was funded by the National Natural Science Foundation of China (Grant No. 62076263), Guangdong Natural Science Funds for Distinguished Young Scholar (Grant No. 2017A030306028), and Guangdong Province Key Laboratory of Big Data Analysis and Processing.

## Author contributions

S.H. proposed the research, conducted experiments, analyzed the data, and wrote the manuscript. H.H.Z. improved the manuscripts and supervised the overall project. G.S. analyzed the experimental results. K.J. improved the manuscripts. Z.Z. improved the manuscripts.

## Competing interests

The authors declare no competing interests.
