## [Peer Review File · Communications Chemistry]

nature portfolio

Peer Review File

Retrosynthetic Planning with Experience-Guided Monte Carlo Tree SearchThis manuscript has previously been reviewed at another Nature Portfolio journal. This document only contains reviewer comments and rebuttal letters for versions considered at Communications Chemistry.

Reviewers' comments:

Reviewer #3 (Remarks to the Author):

Previous concern:

I agree that the authors have addressed most of the concerns, except for the second one. In lines 291-292, the authors still refer to the dataset construction in the appendix, which is critical to understanding the work. I urge the authors to include the construction and filtering of data in the main text.

Additional major concerns:

Description of the main contribution:

Firstly, the description of single and multi-step retrosynthesis is incorrect. All multi-step retrosynthesis relies on single-step retrosynthesis to determine the possible next steps. What the author describes as a "score function" in their MCTS is actually a template-based single-step retrosynthesis model, and the training strategy of the single-step retrosynthesis model is the main contribution of this paper. The authors should compare their work to the reinforcement learning-based approaches (Schreck, Coley, and Bishop 2019; Wang et al. 2020) as a baseline.

Target audience:

The article is written in a format more suitable for a computer science journal/conference than a chemistry one. In Figure 1, the authors hypothesize a case of multiple synthetic pathways of A instead of providing a valid example of a true synthetic pathway. The first footnote introducing what "reaction" and "reaction template" are is written for a computer science background.

Questionable rationale:

The authors state that Retro* is only trained on successfully synthesized molecules such as TA2 but may ignore the potentially better choice of TA1 that was not in the training set. The same statement could apply to EG-MCTS as well. EG-MCTS initially trains its model by experience collecting from a randomly initialized network, which is equivalent to randomly picking branches. I urge the authors to re-analyze what contributes to their empirically better performance.

Unnecessary content:

The second paragraph introducing the previous work on related topics is too long and distracting. Line 58: DINGOS is a method for synthesizable molecular design, not either single-step or multi-step retrosynthesis. The example from lines 97 to 123 is not providing much information, and the authors should summarize the defect in one or two sentences. The paper does not need to state the difference between their method and AlphaGo, and I recommend deleting Appendix (I).

Additional minor concerns:

Line 67, the program is called Synthia now. (<https://www.sigmaaldrich.com/US/en/services/software-and-digital-platforms/synthia-retrosynthesis-software>)

In Figure 3, lines 422-424, the authors state that their route is better because it is one step shorter. However, the "goodness" of synthetic paths also includes factors such as the final yield, ease of purification, selectivity, and environmental friendliness. Retro*+'s route protects the nitrogen in the piperazine, preventing the by-reaction of bi-arylation in this step, and the deprotection can be conducted without purification, i.e., in a one-pot manner. Therefore, Retro*+'s route potentially has a higher yield without more effort. The part comparing the "length" of synthetic pathways should be deleted.

Line 671: the subtitle is incorrect as this paper does not solve the problem of drug design.

Reviewers' comments:

Reviewer #3 (Remarks to the Author):

Previous concern:

I agree that the authors have addressed most of the concerns, except for the second one. In lines 291-292, the authors still refer to the dataset construction in the appendix, which is critical to understanding the work. I urge the authors to include the construction and filtering of data in the main text.

Response #1: We removed the Appendix section and placed the dataset construction and the filtering process in Lines 279 to 313.

Additional major concerns:

Description of the main contribution:

Firstly, the description of single and multi-step retrosynthesis is incorrect. All multi-step retrosynthesis relies on single-step retrosynthesis to determine the possible next steps. What the author describes as a "score function" in their MCTS is actually a template-based single-step retrosynthesis model, and the training strategy of the single-step retrosynthesis model is the main contribution of this paper. The authors should compare their work to the reinforcement learning-based approaches (Schreck, Coley, and Bishop 2019; Wang et al. 2020) as a baseline.

Response #2: We revised the description regarding single and multi-step retrosynthesis in Lines 44 to 49, and the score function in Lines 52 to 55. We also added a few sentences in Lines 55 to 59, Lines 72 to 73 and Lines 89 to 93 to further clarify the idea of the score function.

We would like to clarify that our proposed EGN is not a single-step retrosynthesis model, but a prediction model that learns from multi-step retrosynthetic experience that evaluates the score of templates acting on molecules in the long run.

Regarding the comparison with reinforcement learning-based baselines, we did compare our work to a reinforcement learning-based baseline, i.e., Retro*+ (Kim et al, ICML 2021, c.f. Tables 1 and 2), which used the successful synthetic routes to enhance the probabilities of the corresponding templates based on imitation learning, one of reinforcement learning techniques. For the

mentioned reinforcement learning-based approaches (Schreck, Coley, and Bishop 2019; Wang et al. 2020), we did try to conduct experiments but failed. Although the source Github code was provided by authors, the best performance as mentioned in their paper could not be reproduced based on the code and settings provided in the paper. The same issue was also encountered by other users who tried to run the same Github code provided by authors. We either did not see any trained model provided by authors for users to reproduce the results without training the model. Besides, authors of Retro* mentioned their differences (advantages) from (Schreck, Coley, and Bishop 2019). For the approach proposed by Wang et al. 2020, the authors did not release their experimental code and data.

Target audience:

The article is written in a format more suitable for a computer science journal/conference than a chemistry one. In Figure 1, the authors hypothesize a case of multiple synthetic pathways of A instead of providing a valid example of a true synthetic pathway. The first footnote introducing what "reaction" and "reaction template" are is written for a computer science background.

Response #3: We replaced the example with a real example shown in Figure 1. We also added the corresponding analysis in Lines 94 to 111, Lines 118 to 121 and Lines 165 to 170.

Questionable rationale:

The authors state that Retro* is only trained on successfully synthesized molecules such as TA2 but may ignore the potentially better choice of TA1 that was not in the training set. The same statement could apply to EG-MCTS as well. EG-MCTS initially trains its model by experience collecting from a randomly initialized network, which is equivalent to randomly picking branches. I urge the authors to re-analyze what contributes to their empirically better performance.

Response #4: We added analysis in Lines 151 to 162. The base algorithm of Retro*, A* search, prefers to search molecules with lower synthetic cost, resulting in selecting template T_{A_2} first. Our EG-MCTS is based on MCTS search. Different from A^* search, the core component of MCTS, "upper confidence bound" (UCB), balances the trade-off between exploration of infrequently-visited routes and exploitation of high-value routes. It

makes the composite score of high-value routes to decrease as the number of visits increases. Even though EGN may predict a higher score for ST_{A_2} in the random initial stage, the search will later turn to explore ST_{A_1} due to the decreasing score of ST_{A_2} after multiple visits to ST_{A_2} . Therefore, our {ours} approach will find that template ST_{A_1} leads to a fewer-step route during the MCTS exploration and record the experiences about ST_{A_1} for future exploration.

Unnecessary content:

The second paragraph introducing the previous work on related topics is too long and distracting. Line 58: DINGOS is a method for synthesizable molecular design, not either single-step or multi-step retrosynthesis. The example from lines 97 to 123 is not providing much information, and the authors should summarize the defect in one or two sentences. The paper does not need to state the difference between their method and AlphaGo, and I recommend deleting Appendix (I).

Response #5: Thanks. We removed the irrelevant previous works, the comparison with AlphaGo, and Appendix (I). We also revised the description of the example (in Lines 97 to 123 of the original version) in Lines 94-111.

Additional minor concerns:

Line 67, the program is called Synthia now.

(<https://www.sigmaaldrich.com/US/en/services/software-and-digital-platforms/synthia-retrosynthesis-software>)

Response #6: Thanks. We corrected the program name in Line 66.

In Figure 3, lines 422-424, the authors state that their route is better because it is one step shorter. However, the "goodness" of synthetic paths also includes factors such as the final yield, ease of purification, selectivity, and environmental friendliness. Retro*+'s route protects the nitrogen in the piperazine, preventing the by-reaction of bi-arylation in this step, and the deprotection can be conducted without purification, i.e., in a one-pot manner. Therefore, Retro*+'s route potentially has a higher yield without more effort. The part comparing the "length" of synthetic pathways should be deleted.

Response #7: Thanks. In this paper we indeed do not consider other factors as you mentioned (as we mentioned in the conclusion section 2.7, Lines 805-810, it is one of the limitations that could be further

investigated in the future). We revised the corresponding description in Line 442 to 446 to avoid confusion.

Line 671: the subtitle is incorrect as this paper does not solve the problem of drug design.

Response #8: We corrected the subtitle, as well as the corresponding description in Lines 693-699.

REVIEWERS' COMMENTS:

Reviewer #3 (Remarks to the Author):

Thanks the authors for the detailed response. I think most concerns are addressed now.